# HPV Vaccination in Women with Cervical Intraepithelial Neoplasia Undergoing Excisional Treatment: Insights into Unsolved Questions

**DOI:** 10.3390/vaccines10060887

**Published:** 2022-06-01

**Authors:** Carla Henere, Aureli Torné, Anna Llupià, Marta Aldea, Cristina Martí, Ariel Glickman, Adela Saco, Lorena Marimon, Carolina Manzotti, Natalia Rakislova, Jaume Ordi, Marta del Pino

**Affiliations:** 1Institute Clinic of Gynecology, Obstetrics, and Neonatology, Hospital Clínic, University of Barcelona, 08007 Barcelona, Spain; chenere@fhes.cat (C.H.); atorne@clinic.cat (A.T.); marti@clinic.cat (C.M.); glickman@clinic.cat (A.G.); 2Institut d’Investigacions Biomèdiques August Pi i Sunyer (IDIBAPS), Hospital Clínic, University of Barcelona, 08007 Barcelona, Spain; masaco@clinic.cat; 3Department of Preventive Medicine and Epidemiology, Hospital Clínic, University of Barcelona, 08007 Barcelona, Spain; allupia@clinic.cat (A.L.); maldea@clinic.cat (M.A.); 4Institute for Global Health (ISGlobal), Hospital Clínic, University of Barcelona, 08007 Barcelona, Spain; lmarimon@clinic.cat (L.M.); carolina.manzotti@isglobal.org (C.M.); rakislova@clinic.cat (N.R.); jordi@clinic.cat (J.O.); 5Department of Pathology, Hospital Clínic, University of Barcelona, 08007 Barcelona, Spain

**Keywords:** HPV vaccine, excisional treatment, post-treatment HSIL

## Abstract

Several questions regarding the role of vaccination in women treated for high-grade cervical intraepithelial lesion (HSIL) have not been clarified. One of the main queries is whether the time at which the vaccine is administered (before or after treatment) influences the protection against post-treatment HSIL. A second unanswered question is whether the vaccine has any effect in women with persistent HPV after treatment. We aimed to address these questions in a study of 398 women undergoing excisional treatment from July 2016 to December 2019. Vaccination was funded and offered to all women undergoing treatment. A total of 306 women (76.9%) accepted HPV vaccination (vaccinated group): 113 (36.9%) received the first dose before excision and 193 (63.1%) after the procedure. A total of 92 women (23.1%) refused the vaccine (non-vaccinated group). Women vaccinated before treatment showed a lower rate of post-treatment HSIL compared with non-vaccinated women (0.9% vs. 6.5%; *p* = 0.047). Among women with persistent HPV infection after treatment, those who had received the vaccine showed a lower prevalence of post-treatment HSIL than non-vaccinated women (2.6% vs. 10.5%; *p* = 0.043). In conclusion, this study shows that HPV vaccination before treatment reduces the prevalence of post-treatment HSIL and suggests that vaccination might even benefit women with persistent HPV after treatment.

## 1. Introduction

Women with high-grade squamous intraepithelial lesion (HSIL), the precursor of cervical cancer, usually undergo excisional treatment to remove the lesion and prevent progression [1]. Interestingly, although high-risk human papillomavirus (HPV) clearance is not the goal of the treatment, elimination of the infection is achieved in many of these women [2,3]. However, despite the clearly demonstrated effectiveness of the excisional procedure, persistent/recurrent HSIL has been described in 5–15% of women undergoing cervical excision [2,3,4,5,6]. Post-treatment HSIL may result from persistence of the HPV infection after excision, associated or not with inadequate removal of the cervical lesion (i.e., treatment failure) or with re-infection with a new HPV type [7] leading to the development of a new lesion. Due to this risk of post-treatment HSIL, close follow-up after treatment is mandatory in these women to detect either infection or disease [8,9,10,11].

In the last few years, there has been a growing body of evidence indicating that HPV vaccination in women treated for HSIL could reduce the prevalence of post-treatment lesions [2,12,13,14]. In keeping with these results, our group recently published a real-life study in treated women, showing that vaccination reduced the risk of subsequent HSIL [2]. However, despite these promising results, several questions regarding the role of vaccination in these women have not yet been clarified. One of the main gaps in knowledge is whether the time at which the vaccine is administered (before or after excisional treatment) has an impact in terms of protection against post-treatment infection and lesion. Some studies have suggested that vaccination before treatment may be more beneficial [15], but this has not been confirmed in other series [2,3]. In our setting, the HPV vaccine is publicly funded for women undergoing treatment. The current recommendations of the Ministry of Health of Catalonia (ref. [16]) emphasize that the vaccine should be administered as soon as possible after HSIL diagnosis, either before or after treatment. Consequently, some women receive the first HPV vaccine dose before and others after excisional treatment.

Another unanswered question is whether the vaccine benefits women with persistent infection after treatment. In our previous study, the benefit of the vaccine was particularly clear in women who cleared HPV infection after treatment [2], whereas it was not statistically significant in women with persistent HPV infection after treatment. However, these last groups of women also showed lower rates of persistent/recurrent disease when vaccinated [2], a finding also observed in a few other studies. It has been proposed that the possible mechanism explaining the benefit of the vaccine in treated women is its protection against re-infection or re-activation of HPV [11,17,18]. Thus, the effect the vaccine may have in women with persistent infection after treatment is not clear, and more evidence is required.

The main objective of the present study is to address the possible effect of the timing of vaccination (before or after excisional treatment) on the protection against post-treatment HSIL. As a secondary objective, we aimed to analyse the effect of vaccination on reducing post-treatment lesions in patients with persistent HPV infection after HSIL excisional treatment. A subset of women who refused vaccination in the same period of recruitment was used as a control group to compare the effectiveness of vaccination.

## 2. Materials and Methods

### 2.1. Selection Criteria

All women undergoing excisional treatment (loop electrosurgical excision procedure) in our institution between July 2016 and December 2019 were eligible for the study. Following the recommendations of the American Society for Colposcopy and Cervical Pathology (ASCCP) [11] and the Spanish Society of Cervical Pathology and Colposcopy (AEPCC) [19], the criteria for treatment were: (1) histological diagnosis of HSIL; and (2) repeated cytological result of HSIL in at least two Pap smears separated by 12 months in patients without histological confirmation of HSIL (cytological-histological discordance). The exclusion criteria for the present study were: (a) ongoing pregnancy; (b) immunosuppression; (c) multicentric HPV disease; (d) previous excisional treatment for cervical lesion; and (e) history of any HPV-associated cancer (uterine cervix, vagina, vulva, anus or head and neck region) or diagnosis of invasive disease in the surgical specimen.

### 2.2. Liquid-Based Cytology and HPV Testing

Cervical samples were collected using a cytobrush and stored in PreservCyt solution (Hologic, Malborough, MA, USA) for liquid-based cytology and HPV testing. A ThinPrep T2000 slide processor (Hologic) was used to prepare thin-layer cytology slides that were stained using the Papanicolaou method. The cytology slides were evaluated following the Bethesda system [20].

For HPV testing, the Cobas HPV test (Cobas 4800, Roche Molecular Diagnostics), based on a real-time polymerase chain reaction (PCR) system, was used. This method detects 14 high-risk HPV types and provides specific genotyping for HPV16 and HPV18.

### 2.3. Pre-Treatment Histological Diagnosis

Colposcopy-directed biopsies and endocervical curettages were fixed in formalin and embedded in paraffin. Four μm sections were stained with haematoxylin and eosin (H and E). In all cases, p16 immunohistochemistry (CINtec histology kit; clone E6H4; mtm-Roche Laboratories, Heidelberg, Germany) was routinely performed following the manufacturer’s protocol. Immunohistochemical staining was performed with the automated BenchMark ULTRA platform (Ventana, Tucson, AZ, USA). Only cases with continuous, “block” staining of cells of the basal and parabasal layers were considered positive [21]. The histological diagnosis was established based on the combination of the H and E findings and the p16-stained sections. Biopsy specimens were classified as negative for SIL, low-grade (L)SIL, or HSIL according to the WHO criteria [22]. The p16 positive staining was considered as a mandatory criterion for the diagnosis of histological HSIL.

### 2.4. Excisional Treatment and Histological Diagnosis of the Surgical Specimen

After colposcopic examination with acetic acid and iodine solution to identify the abnormal area, 1 mL of 1% mupivacain was injected into each quadrant of the cervix. Loop size was selected depending on the size of the area to be excised. The excision was performed under colposcopic vision [3]. When endocervical involvement was suspected (colposcopically abnormal area or transformation zone not completely visible), a second selective endocervical sweep was performed using a smaller loop (top hat). After excision, selective coagulation of the bleeding areas was performed by ball diathermy.

Surgical specimens were anatomically oriented, pinned to a cork support and fixed in 10% neutral buffered formalin. The excisional samples were processed as previously described Briefly, 4μm sections were stained with H and E and examined in toto. The histological diagnosis was established based on the combination of the H and E findings and the p16-stained sections. Only cases showing continuous “block” staining of the basal and parabasal layers were considered positive for p16 [21]. Surgical specimens were classified as negative for SIL, low-grade (L)SIL, or HSIL according to the WHO criteria [22]. The p16 positive staining was considered as a mandatory criterion for the histological diagnosis of HSIL. Surgical margins were identified with ink and carefully examined. Margins were considered negative if no SIL was detected and positive when SIL of any grade was present. In the latter situation, the site (exocervical/endocervical) of involvement was reported.

### 2.5. Post-Treatment Follow-Up

Following current guidelines [11,19,23,24], the first follow-up control was scheduled at 4 months after treatment for women with positive margins in the conization specimen and at 6 months for women with negative margins. In this visit, cervical cytology and HPV testing were performed. If all tests were negative, a second follow-up visit was scheduled 12 months after the first control and included cervical cytology and HPV testing. If one of these tests was positive (abnormal cytology or positive HPV testing), a new colposcopy was performed with a colposcopy-directed biopsy or endocervical curettage, when indicated.

All women with histological diagnosis of HSIL in a colposcopy-directed biopsy and/or endocervical curettage performed in any of the follow-up visits were referred for a second conization.

The mean time from treatment to the first follow-up visit was 6.1 months (standard deviation [SD] 2.1). The mean time of the follow-up to the last control was 20.2 months (SD 10.6).

### 2.6. HPV Vaccination Policy

In July 2017, the Ministry of Health of Catalonia started funding HPV vaccination for women treated in the previous year (from July 2016) [16]. Thus, following government legislation, all women who had undergone excisional treatment due to HSIL in the previous 12 months (from July 2016) and had not been previously vaccinated were called, and free vaccination was offered. Moreover, from July 2017 in our institution, once excisional treatment is indicated, HPV vaccination is offered, and women are referred to the adult immunization department for free administration of the nine-valent (9v) vaccine. The first dose of the vaccine is provided either immediately before or after the treatment, according to the availability of the vaccine and the timing for HSIL treatment. Vaccine administration follows the manufacturer’s recommendations (0, 2, and 6 months). The date of each vaccine dose is recorded.

For the present study, all women who accepted HPV vaccination were included in the study as vaccinated women. Women who refused the vaccine were considered as non-vaccinated women and were included as the control group.

### 2.7. Clinical Outcomes and Definitions

The clinical outcomes of the patients, either at the first post-treatment control (at 4/6 months after conization) or at the end of the follow-up, were categorized as follows: (1) post-treatment HSIL (presence of histologically confirmed HSIL, or a repeated HSIL result in at least 2 PAP smears separated by 6 months and positive HPV testing results, independently of the histological diagnosis); (2) post-treatment HPV (positive HPV testing with either HSIL cytology in one Pap smear or <HSIL Pap smear result, and biopsy diagnosis, if any, showing LSIL or no disease); and (3) no evidence of disease (negative HPV testing, negative cervical cytology, and, if available, negative biopsy).

As there is no consensus for the definitions of SIL persistence vs. recurrence and/or HPV persistence vs. recurrence after treatment [7,25,26], in this study the terms post-treatment HSIL and post-treatment HPV include the two conditions (persistence and recurrence).

The end of the follow-up was defined as the time to the diagnosis of post-treatment HSIL or to the last recorded visit.

### 2.8. Data Analysis

Data were analysed with SPSS software (version 28.0.0.0, SPSS, Inc., Chicago, IL, USA). A sample size calculation was not performed. However, to achieve the highest sample size possible, all women undergoing excisional treatment in our institution between July 2016 and December 2019 were included in the study.

All women with histologically confirmed HSIL after treatment were considered as having developed post-treatment HSIL. As post-treatment HSIL was one of the main outcomes of the study for patients with persistent/recurrent disease, the follow-up of the women undergoing a new excisional treatment after the second procedure was disregarded for the purposes of the study. Comparisons were performed considering as outcome either “clinical outcome” (post-treatment HSIL vs. post-treatment HPV vs. no evidence of disease) or specifically “post-treatment HSIL” (vs. no post-treatment HSIL). The results for the vaccinated and non-vaccinated women were compared. The time of vaccination (before or after excisional treatment) was also analysed. HPV genotypes were considered as HPV16 and/or HPV18 and other HPV types, including infections by high-risk HPV different from 16 and 18.

Categorical variables are presented as absolute numbers and percentages and compared using the χ^2^ or Fisher exact test. Continuous variables are presented as mean and standard deviation (SD). Means were compared using the analysis of variance test.

## 3. Results

### 3.1. Clinical Characteristics of the Vaccinated and Non-Vaccinated Women

During the study period, a total of 523 women were treated for SIL in our institution and were therefore eligible. Nine women were excluded from the study because they were pregnant at the moment of HSIL diagnosis. Another 28 women were excluded from the study because of immunosuppression conditions (12 women living with HIV; 10 organ or stem cell transplantation recipients, and 6 patients with chronic diseases treated with high doses of corticoids). Finally, 42 women with multicentric disease, 40 women who had undergone previous treatment for SIL/cervical intraepithelial neoplasia (CIN), and 6 with early-stage cervical cancer in the surgical specimen were also excluded from the analysis. Thus, a total of 398 patients were included in the study.

The mean age of the 398 women included in the study was 39.8 years (SD 10.1). A total of 306 women (76.9%) accepted vaccination and received at least one dose of the HPV vaccine (vaccinated group). Of the 306 vaccinated women, 281 (91.8%) received three doses, 19 (6.2%) received two doses, and 6 (2.0%) received only one dose. A total of 92 women (23.1%) refused the vaccine (non-vaccinated group). Table 1 shows the baseline characteristics of the two groups of women (vaccinated and non-vaccinated women) as well as the final diagnoses of the surgical specimens. Of the cohort, 317 (79.6%) women showed an HSIL lesion in the surgical specimen, 40 (10.1%) women had LSIL, and 41 (10.3%) did not show any lesion.

### 3.2. Vaccination Schedule and Cervical Treatment

Among the 306 vaccinated women, 113 (36.9%) received the first vaccine dose before excision and 193 (63.1%) after the procedure. The mean time between the first dose and excisional treatment was 4.1 months (SD 11.1) for women receiving the vaccine before and 5.0 months (SD 3.9) for women vaccinated after treatment (*p* = 0.364). No differences were found between the women receiving the vaccine before and those who were vaccinated after the treatment in terms of mean age (38.2 vs. 39.8, respectively; *p* = 0.132), histological result of the conization specimen (HSIL 81.4% vs. 85.5%; LSIL 8.8% vs. 6.2%; negative 9.7% vs. 8,3; *p* = 0.608), positive margin status (36.3% vs. 40.1%; *p* = 0.277), or number of doses of the HPV vaccine (three doses 94.7% vs. 90.2%; two doses 3.5% vs. 7.8%; one dose 1.8% vs. 2.1; *p* = 0.325).

### 3.3. Clinical Outcome in the First Control after Treatment

Table 2 shows the relation between the pre-treatment HPV genotype, the indication of treatment, surgical specimen diagnosis, margin status and vaccination status, and the clinical outcome at the first follow-up control. Margin status and vaccination status were associated with the clinical outcome in the first follow-up control.

#### Post-Treatment HSIL in the First Control after Treatment According to Vaccination Status

The prevalence of post-treatment HSIL at the first follow-up control was significantly lower in women who received the vaccine before treatment compared with the group of women vaccinated after treatment (1/113 [0.9%] vs. 4/193 [2.1%]; *p* = 0.040) and the group of non-vaccinated women (4/92 [4.3%], *p* = 0.037). No differences were identified in terms of the mean time to post-treatment HSIL diagnosis at the first follow-up control (7.2 months [SD 3.4]) for the vaccinated women and 5.2 months [SD 1.5] for the non-vaccinated women (*p* = 0.331).

No significant differences were found in the prevalence of post-treatment HSIL between the women vaccinated after treatment and the non-vaccinated women (*p* = 0.553). The nine women with post-treatment HSIL at the first follow-up control were referred for a second conization. The procedure was not performed in three of these nine women. Two of these three women did not show any lesion in the colposcopy evaluation the day of the conization, and therefore, the procedure was cancelled. None showed evidence of disease during the follow-up. One woman rejected the second conization and showed persistent HPV infection without cervical lesion in the following controls.

### 3.4. Clinical Outcome at the End of the Follow-Up

Table 3 shows the relation between the pre-treatment HPV genotype, indication of treatment, surgical specimen diagnosis, margin status, vaccination status, and the clinical outcome at the final follow-up control. None of the variables were associated with the outcome at the end of the follow-up.

None of the women with no evidence of disease in the first follow-up control at six months showed post-treatment HSIL in the last follow-up visit. In contrast, 4.6% [7/153] of the women with positive post-treatment HPV testing and 55.6% [5/9] of the women with HSIL in the first follow-up showed post-treatment HSIL at the end of the follow-up (*p* < 0.001).

No differences were observed in terms of the prevalence of HPV16/18 at the end of the follow-up between women vaccinated before or after treatment and non-vaccinated women (8.8% [10/113] vs. 5.7% [11/193] vs. 6.5% [6/92]; *p* = 0.795).

#### Post-Treatment HSIL at the End of the Follow-Up According to Vaccination Status

The overall group of vaccinated women showed a lower prevalence of post-treatment HSIL than non-vaccinated women (1.9% [6/306], vs. 6.5% [6/92]; *p* = 0.037). No differences were identified in terms of the mean time to post-treatment HSIL diagnosis (18.9 months [SD 12.4] for the vaccinated women and 28.4 months [SD 19.0] for the non-vaccinated women, *p* = 0.330). Women vaccinated before treatment showed a lower rate of post-treatment HSIL compared to non-vaccinated women (0.9% [1/113] vs. 6.5% [6/92]; *p* = 0.047), whereas no significant differences were found compared with women vaccinated after treatment (2.6% [5/193]; *p* = 0.299). Neither were there were any statistically significant differences in the rate of post-treatment HSIL between women vaccinated after treatment and the non-vaccinated women (2.6% [5/193) vs. 6.5% [6/92]; *p* = 0.272).

Vaccinated women who received two or three doses showed a lower risk of post-treatment HSIL at the end of the follow-up compared with women who received only one dose (1.7% [5/300] vs. 16.7% [1/6]; *p* = 0.028) or non-vaccinated women (6.5% [6/92]; *p* = 0.014). No differences were found in terms of post-treatment HSIL between women vaccinated with three doses compared with the group of women vaccinated with two doses (1.8% [5/281] vs. 0.0% [0/19]; *p* = 0.558).

Table 4 shows the clinical features, the results at the baseline visit and at the first post-treatment control, as well as the vaccination scheme and the HPV genotype at the end of the follow-up in the vaccinated and unvaccinated women who developed HSIL at the end of the follow-up. The pre- and post-treatment HPV genotype in women who developed post-treatment HSIL were concordant in 11 out of the 12 women.

### 3.5. Effect of Vaccination in Women with Persistent HPV Infection after Treatment

Among women with positive HPV infection in the first follow-up control at 4/6 months post-treatment (*n* = 153), the vaccinated women showed a lower prevalence of post-treatment HSIL at the end of the follow-up than non-vaccinated women (2.6% [3/115] vs. 10.5% [4/38]; *p* = 0.043). The prevalence of post-treatment HSIL at the end of the follow-up, was similar in all vaccinated women, independently of the time of vaccination (2.4% [2/82] for those vaccinated before treatment vs. 3.0% [1/33] women vaccinated after treatment; *p* = 0.857).

## 4. Discussion

The present series suggests that vaccination before excisional treatment is more effective than vaccination after treatment. Remarkably, although women vaccinated after the excisional treatment also showed a lower prevalence of post-treatment HSIL compared with non-vaccinated women, the differences did not reach statistical significance. These findings are in keeping with a recent study conducted in Denmark including more than 17,000 women undergoing excisional treatment due to HSIL [15]. This study concluded that women vaccinated before conization (between 0 and 3 months) had a lower absolute risk of HSIL during the follow-up compared to non-vaccinated women. Interestingly, the risk of HSIL in women vaccinated after conization (0–12 months) was similar to that of non-vaccinated women [15], which is in keeping with the findings of the present study. It has been suggested that vaccination before treatment may ensure having a sufficient amount of anti-HPV neutralising antibodies in the cervicovaginal area at the time of excision (removing most of the infected cells), which may prevent re-infection of the basal layer cells [27].

In recent years, several studies have evaluated the efficacy of shorter vaccine schedules in adolescents [28,29,30,31,32], and two HPV vaccine doses is already the standard schedule in young girls. There is also preliminary evidence suggesting that two-dose schemes might also be applicable to adult women [31,32]. In the present series, similar results were observed in women who received two or three doses, who showed a lower rate of HSIL at the end of the follow-up than women who had received only one dose or had not been vaccinated. However, this study was not designed to compare the efficacy of one or more doses of the HPV vaccine, a problem that would need studies with an appropriate design.

The prevalence of positive margins in the surgical specimen was 38.2% in the present study. Previous studies performed in non-vaccinated women have shown that positive margins represent a risk factor for developing persistent/recurrent disease [18,33]. Interestingly, in our study, all vaccinated women, including those with positive margins, showed a very low rate of post-treatment HSIL, suggesting that the vaccine might have a protective role even in this high-risk subgroup of treated women. These results were in keeping with our previous study in women undergoing cervical treatment [2].

Surprisingly, although the differences in the prevalence of post-treatment HSIL were evident, no differences were observed in terms of the prevalence of HPV16/18 at the end of the follow-up between women vaccinated before or after treatment and non-vaccinated women (8.8% vs. 5.7% and 6.5%, respectively). This finding suggests that the benefit of vaccination in women undergoing treatment might not only rely on the prevention of acquiring new infections of the HPV genotypes included in the vaccine. In the present series, the HPV 16/18 genotype was not related to the outcome. This is in keeping with a previous study published by our group, which included a different subset of women [2].

In our previous study the HPV vaccine showed a non-significant protective effect against post-treatment HSIL in women with HPV infection in the first control (scheduled at six months after treatment) [2]. In the present series, which included a larger number of patients and involved a different cohort of women, vaccinated women with a post-treatment HPV positive test in the first control showed a significantly lower prevalence of post-treatment HSIL at the end of the follow-up compared with non-vaccinated women. Few previous series have reported benefits of the HPV vaccine in women with prevalent HPV infection, including women treated for HSIL [2,12,34,35]. Interestingly, a recent randomised controlled trial including 312 women with SIL persistence after conservative cervical treatment reported the overall efficacy of the vaccine, defined as lesion regression of greater than 50% in women with persistent LSIL or HSIL [36].

The main strength of our study is that all the participants were intensively screened, with rigorous assessment of disease end points and procedures. However, there are some limitations that should be noted. The main limitation is that women were vaccinated at different time points, and due to the real-life nature of the study, the time between vaccination and treatment varied from woman to woman. Thus, the cumulative risk of developing post-treatment HSIL had to be evaluated by looking at the prevalence of disease, as regression models could not be applied because the assumption of proportional hazards was not fulfilled. Another limitation is that the HPV testing method did not allow extensive HPV genotyping; thus, the effectiveness of the vaccine for individual HPV genotypes could not be properly assessed. However, women with post-treatment HSIL and positive HPV testing at the end of the follow-up showed high concordance between the pre- and post-treatment HPV genotype, suggesting that HPV persistence might be an important issue related to the risk of subsequent disease after treatment in adult women.

## 5. Conclusions

The present series provides further evidence confirming the reduction of post-treatment HSIL in patients undergoing cervical treatment who receive HPV vaccination. Moreover, although the differences between the two strategies of vaccine administration (pre- and post-treatment) did not reach statistical significance, the results of the present series suggest that pre-treatment vaccination confers better protection. Additionally, we found that vaccination might even offer benefits in women with persistent HPV after treatment. Further analysis involving a larger number of patients is needed to confirm these answers to the unsolved questions regarding the time of vaccination and its effect in treated women with persistent HPV infection.

## Figures and Tables

**Table 1 vaccines-10-00887-t001:** Epidemiological characteristics at the baseline visit and pathological results of the vaccinated and non-vaccinated women. Values are presented as mean ± standard deviation or absolute numbers and percentages.

	Vaccinated Women (*n* = 306)		Non-Vaccinated Women (*n* = 92)		
	*n*	(%)	*n*	(%)	*p* Value *
**Age**	39.2	±9.1	41.8	±12.8	0.071
**Smoking habit**					0.850
Active smoker	122	(39.9)	37	(40.2)	
Ex-smoker	40	(13.1)	10	(10.9)	
Non-smoker	105	(34.3)	33	(35.9)	
No data	39	(12.7)	12	(13.0)	
**HPV genotype (pre-treatment)**					0.103
HPV16 and/or HPV18	191	(62.4)	46	(50.0)	
Other HPV types	108	(35.3)	43	(46.7)	
HPV negative	7	(2.7)	3	(3.3)	
**Indication of treatment**					<0.001
Histological HSIL	285	(93.1)	73	(79.3)	
HSIL in two Pap smears	21	(6.9)	19	(20.7)	
**Surgical specimen diagnosis**					<0.001
HSIL	257	(84.0)	60	(65.2)	
LSIL	22	(7.2)	18	(19.6)	
Negative	27	(8.8)	14	(15.2)	
**Margin of surgical specimen**					0.346
Negative	187	(61.1)	59	(64.1)	
Positive	119	(38.9)	33	(35.9)	

HPV: high-risk human papillomavirus; other HPV: high-risk HPV types other than 16 or 18; LSIL: low-grade squamous intraepithelial lesion; HSIL: high-grade squamous intraepithelial lesion. * Fisher exact test.

**Table 2 vaccines-10-00887-t002:** Association between pre-treatment HPV genotype, indication of treatment, surgical specimen diagnosis, margin status, and vaccination status with the outcome at the first follow-up control at 4/6 months post-treatment. Values are presented as absolute numbers and percentages.

	Clinical Outcome in the First Follow-Up Control						
	Post-Treatment HSIL (*n* = 9)		Post-Treatment HPV (*n* = 153)		No Evidence of Disease (*n* = 236)		
	*n*	(%)	*n*	(%)	*n*	(%)	*p* Value *
**HPV genotype (pre-treatment)**							0.933
HPV16 and/or HPV18	6	(2.5)	90	(38.0)	141	(59.5)	
Other HPV types	3	(2.0)	60	(39.7)	88	(58.3)	
HPV negative	0	(0.0)	3	(30.0)	7	(70.0)	
**Indication of treatment**							0.273
Histological HSIL	8	(2.2)	133	(37.2)	217	(60.6)	
HSIL in two Pap smears	1	(2.5)	20	(50.0)	19	(47.5)	
**Diagnosis of surgical specimen**							0.546
HSIL	8	(2.5)	127	(40.1)	182	(57.4)	
LSIL	1	(2.5)	13	(32.5)	26	(65.0)	
Negative	0	(0.0)	13	(31.7)	28	(68.3)	
**Margin status of surgical specimen**							0.006
Negative	3	(1.2)	83	(33.7)	160	(65.1)	
Positive	6	(3.9)	70	(46.1)	76	(50.0)	
**Vaccination status**							0.050
Before treatment	1	(0.9)	33	(29.2)	79	(69.9)	
After treatment	4	(2.1)	82	(42.5)	107	(55.4)	
Non-vaccinated women	4	(4.3)	38	(41.3)	50	(54.4)	

HPV: high-risk human papillomavirus; other HPV: high-risk HPV types other than 16 or 18; LSIL: low-grade squamous intraepithelial lesion; HSIL: high-grade squamous intraepithelial lesion. * Fisher exact test.

**Table 3 vaccines-10-00887-t003:** Association between pre-treatment HPV genotype, indication of treatment, surgical specimen diagnosis, margin status, and vaccination status with the outcome at the end of the follow-up. Values are presented as absolute numbers and percentages.

	Clinical Outcome at the End of Follow-Up						
	Post-Treatment HSIL (*n* = 12)		Post-Treatment HPV (*n* = 75)		No Evidence of Disease (*n* = 311)		
	*n*	(%)	*n*	(%)	*n*	(%)	*p* Value *
**HPV genotype (pre-treatment)**							0.568
HPV16 and/or HPV18	5	(2.1)	46	(19.4)	186	(78.5)	
Other HPV types	7	(4.6)	28	(18.6)	116	(76.8)	
HPV negative	0	(0.0)	1	(10.0)	9	(90.0)	
**Indication of treatment**							0.105
Histological HSIL	10	(2.8)	63	(17.6)	285	(79.6)	
HSIL in two Pap smears	2	(5.0)	12	(30.0)	26	(65.0)	
**Diagnosis of surgical specimen**							0.314
HSIL	11	(3.5)	54	(17.0)	252	(79.5)	
LSIL	1	(2.5)	11	(27.5)	28	(70.0)	
Negative	0	(0.0)	10	(24.4)	31	(75.6)	
**Margin status of surgical specimen**							0.337
Negative	5	(2.0)	46	(18.7)	195	(79.3)	
Positive	7	(4.6)	29	(19.1)	116	(76.3)	
**Vaccination status**							0.199
Before treatment	1	(0.9)	20	(17.7)	92	(81.4)	
After treatment	5	(2.6)	38	(19.7)	150	(77.7)	
Non-vaccinated women	6	(6.5)	17	(18.5)	69	(75.0)	

HPV: high-risk human papillomavirus; other HPV: high-risk HPV types other than 16 or 18; LSIL: low-grade squamous intraepithelial lesion; HSIL: high-grade squamous intraepithelial lesion. * Fisher exact test.

**Table 4 vaccines-10-00887-t004:** Clinical characteristics, pre-treatment tests results, diagnosis of surgical specimen, first post-treatment control outcome, HPV genotype at the end of the follow-up, and vaccination scheme in women presenting post-treatment HSIL at the end of the follow-up.

Case	Age	Pre-Treatment HPV Genotype	Indication of Treatment	Diagnosis of Surgical Specimen	Margin Status	First Post-Treatment Control (6 m)	HPV Genotype at the End of Follow-Up	Vaccination Scheme	Doses	Time to Post-Treatment HSIL
**Vaccinated women presenting post-treatment HSIL (*n* = 6)**										
1	45.7	Other HPV	Histological HSIL	HSIL	Negative	HSIL	Other HPV	After treatment	3	8.8
2	41.7	Other HPV	Histological HSIL	HSIL	Positive	HPV	Other HPV	After treatment	3	35.9
3	48.1	HPV 16/18	Histological HSIL	HSIL	Positive	HSIL	HPV 16/18	After treatment	3	6.7
4	48.3	HPV 16/18	Histological HSIL	HSIL	Negative	HPV	HPV 16/18	Before treatment	3	23.2
5	35.6	Other HPV	Histological HSIL	HSIL	Positive	HPV	Other HPV	After treatment	1	15.0
6	42.8	HPV 16/18	Histological HSIL	HSIL	Positive	HSIL	HPV 16/18	After treatment	3	4.2
**Non-vaccinated women presenting post-treatment HSIL (*n* = 6)**										
7	68.0	Other HPV	HSIL in two Pap smears	HSIL	Positive	HPV	Other HPV	-	-	6.5
8	33.0	Other HPV	Histological HSIL	HSIL	Positive	HSIL	Other HPV	-	-	46.1
9	55.1	HPV 16/18	HSIL in two Pap smears	LSIL	Negative	HSIL	HPV 16/18	-	-	6.3
10	26.9	HPV 16/18	Histological HSIL	HSIL	Positive	HPV	Negative	-	-	10.3
11	56.5	Other HPV	Histological HSIL	HSIL	Negative	HPV	Other HPV	-	-	54.2
12	57.1	Other HPV	Histological HSIL	HSIL	Negative	HPV	Other HPV	-	-	25.4

HPV: high-risk human papillomavirus; other HPV: high-risk HPV different than 16 or 18; LSIL: low-grade squamous intraepithelial lesion; HSIL: high-grade squamous intraepithelial lesion.

## Data Availability

Subject numbers were sequentially assigned to the women enrolled in the study. All data collected in this research protocol were treated as confidential and identified with the woman’s study number and not with the woman’s name or address. The cervical cytology results and the biopsy and LEEP results were recorded in the registry of histopathology and cytopathology as well as in the patients’ clinical file. All study records were archived in a safe and secure location. To verify the accuracy of the data, these records are available to the representatives of the national government (e.g., Inspection of Public Health) and licensed inspectors of foreign governments. Members of the Medical Ethical committee are allowed to inspect the quality of the accomplished research. All study data and human materials will be kept for 15 years.

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
