# Peer review of "HPV Vaccination in Women with Cervical Intraepithelial Neoplasia Undergoing Excisional Treatment: Insights into Unsolved Questions"

_vaccines, 2022, doi:10.3390/vaccines10060887_

Round 1
Reviewer 1 Report
The manuscript reports data on the clinical outcome of the excisional treatment of CIN2/3 lesions, in relation to the administration of HPV vaccination and some clinico-virological parameters. The main findings are the protective effect of the vaccination (statistically significant) and a statistically non significant higher efficacy when the vaccine is administred before (compared to after) treatment. The protective effect of the vaccination is in line with previous reports, while the timing of vaccine administration (before or after treatment) is still an unsolved question. The study provides useful information; since it is an analysis of routine data, it gives an insight into the real life and has some limitations, that in my opinion can be reduced by adding additional informations.
MAJOR COMMENTS:
-MATERIALS AND METHODS: in the paragraph 2.4, the histological diagnoses of the excised lesions should be added; in the paragraph 2.5, the criteria for performing a second excision in women with persistent HSIL during follow-up should be provided; in paragraph 2.8, please specify how the women who underwent a second ecxision have been considered (they should have been excluded from further analyses or at least distinguished from the other women).
-RESULTS: please, provide more data on the number and timing of persistent/recurring lesions, distinguishing lesion's grade (CIN2/3 vs CIN1). The pre-treatment HPV typing (HPV16 and/or HPV18 vs other HPV types) distribution showed no correlation with the clinical outcome; since in Table 3, HPV16/18 were more frequent among women without evidence of disease than among those with post-treatment HSIL, further analyses could be done. For the 12 women presenting post-treatment HSIL and detailed in Table 4, to correlate the HPV genotype detected pre-treatment and at the end of follow-up, it is necessary to know the time during follow-up post-treatment HSIL was detected and how is was managed, particularly who of the women underwent a second excision, since this can have modified the natural history. Finally, did the authors evaluate the outcome separately for CIN2 and CIN3?
-DISCUSSION: the paragraph on the margins' status and its correlation with the clinical outcome could be implemented, taking into account the findings of the paper published in BJOG 2020;127:377-387.
-ABSTRACT: the concluding sentence declaring that "...HPV vaccination before treatment reduces the prevalence..." overstates the results, that are not reported in the abstract itself; I would suggest to give more room to this aspect, reporting that the difference between pre- and post-treatment vaccine administration is statistically not significant, although it is suggestive of better protection conferred by pre-treatment vaccination.
Author Response
REVIEWER 1
QUESTION 1:
MATERIALS AND METHODS: in the paragraph 2.4, the histological diagnoses of the excised lesions should be added;
RESPONSE:
We agree with the reviewer that this is important information to be described. Thus, instead of referring the reader to the previous section (“2.3. Pre-Treatment Histological Diagnosis”), where the histological diagnosis was described, we have modified the paragraph in section 2.4 to include a brief description of the procedure, as recommended by the reviewer.
The new version of the manuscript (page 3, lines 117-125, crossed out the deleted text and underlined the new information) now reads:
“Surgical specimens were anatomically oriented, pinned to a cork support and fixed in 10% neutral buffered formalin. The excisional samples were processed as previously described. Briefly, 4μm sections were stained with H&E and examined in toto. The histological diagnosis was established based on the combination of the H&E findings and the p16-stained sections. Only cases showing continuous “block” staining of the basal and parabasal layers were considered positive for p16 (21). Surgical specimens were classified as negative for SIL, low-grade (L)SIL, or HSIL according to the WHO criteria (22). p16 positive staining was considered as a mandatory criterion for the histological diagnosis of HSIL., (see previous section) and examined in toto.”
QUESTION 2:
MATERIALS AND METHODS: in the paragraph 2.5, the criteria for performing a second excision in women with persistent HSIL during follow-up should be provided
RESPONSE:
Following the reviewer’s suggestion, we have added a new sentence to the 2.5 section to define the criteria for performing a second excision in women with persistent HSIL during follow-up. The revised version (page 3, lines 132-140, underlined the new sentence) now reads:
“In this visit, cervical cytology and HPV testing were performed. If all tests were negative, a second follow-up visit was scheduled 12 months after the first control and included cervical cytology and HPV testing. If one of these tests were positive (abnormal cytology or positive HPV testing) a new colposcopy was performed with colposcopy-directed biopsy or endocervical curettage, when indicated. All women with histological diagnosis of HSIL in a colposcopy-directed biopsy and/or endocervical curettage performed in any of the follow-up visits were referred for a second conization.”
QUESTION 3:
MATERIALS AND METHODS: in paragraph 2.8, please specify how the women who underwent a second excision have been considered (they should have been excluded from further analyses or at least distinguished from the other women).
RESPONSE:
Following the reviewer’s recommendation a new sentence has been included in the 2.8 section to clarify this issue.
The revised version (page 4, lines 178-186, underlined the new sentence) now reads:
Data were analyzed with SPSS software (version 28.0.0.0, SPSS, Inc, Chicago, IL, USA). A sample size calculation was not performed. In contrast, to include the highest sample size possible, all women undergoing excisional treatment in our institution between July 2016 and December 2019 were considered for the study. A sample size calculation was not performed. However, to reach the highest sample size possible, all women undergoing excisional treatment in our institution between July 2016 and December 2019 were included in the study. Women undergoing a second conization for histological HSIL during follow-up were considered as post-treatment HSIL and excluded from further analysis. Comparisons were performed considering as outcome either “clinical outcome” (post-treatment HSIL vs. post-treatment HPV vs. no evidence of disease) or specifically “post-treatment HSIL” (vs. no post-treatment HSIL).
Moreover, a new paragraph has been added to the results section (3.3.1. Post-Treatment HSIL in the First Control after Treatment According to Vaccination Status) to further explain the management of women with post-treatment HSIL in the first follow-up control who were referred for a second conizaton
The revised version (page 7, lines 249-256, underlined the new paragraph) now reads:
No significant differences were found in the prevalence of post-treatment HSIL between the women vaccinated after treatment and the non-vaccinated women (p=0.553).
The nine women with post-treatment HSIL at the first follow-up control were referred for a second conization. The procedure was not performed in three of these nine women. Two of these three women did not show any lesion in the colposcopy evaluation the day of the conization and, therefore, the procedure was cancelled. None showed evidence of disease during follow-up. One woman rejected the second conization and showed persistent HPV infection without cervical lesion in the following controls.
QUESTION 4:
RESULTS: please, provide more data on the number and timing of persistent/recurring lesions, distinguishing lesion's grade (CIN2/3 vs CIN1).
RESPONSE:
Data on persistent/recurrent lesions during follow-up were provided in Table 2 (Clinical outcome at first follow-up control) and Table 3 (Clinical outcome at the end of follow-up). We considered that repeating the numbers in the text would be redundant. However, we agree with the reviewer that data on timing of post-treatment lesions is relevant. Therefore, following the reviewer’s suggestion a new sentence has been added to provide this information in the 3.3.1 section (Post-Treatment HSIL in the First Control after Treatment According to Vaccination Status; page 7, lines 242-248, underlined the new text):
The prevalence of post-treatment HSIL at the first follow-up control was significantly lower in women who received the vaccine before treatment compared with the group of women vaccinated after treatment (1/113 [0.9%] vs. 4/193 [2.1%]; p 0.040) and the group of non-vaccinated women (4/92 [4.3%], p=0.037). No differences were identified in terms of the mean time to post-treatment HSIL diagnosis at the first follow-up control (7.2 months [SD 3.4] for the vaccinated women and 5.2 months [SD 1.5] for the non-vaccinated women (p = 0.331).
and 3.4.1 section (Post-Treatment HSIL at the End of the Follow-Up According to Vaccination Status; page 8, lines 277-281, underlining the new text):
The overall group of vaccinated women showed a lower prevalence of post-treatment HSIL than non-vaccinated women (1.9% [6/306], vs. 6.5% [6/92]; p=0.037). No differences were identified in terms of the mean time to post-treatment HSIL diagnosis (18.9 months [SD 12.4] for the vaccinated women and 28.4 months [SD 19.0] for the non-vaccinated women, p = 0.330).
Finally, we agree with the reviewer that distinguishing between high-grade (CIN2/3) and low-grade lesions (CIN1) is very important. In the manuscript CIN2/3 and CIN1 are already considered as two different outcomes as described in 2.7 section (Clinical Outcomes and Definitions). CIN2/3 are included in the group of post-treatment HSIL while CIN1 lesions are included as post-treatment HPV.
QUESTION 5:
RESULTS: The pre-treatment HPV typing (HPV16 and/or HPV18 vs other HPV types) distribution showed no correlation with the clinical outcome; since in Table 3, HPV16/18 were more frequent among women without evidence of disease than among those with post-treatment HSIL, further analyses could be done.
RESPONSE:
Although, as pointed out by the reviewer, the prevalence of HPV 16/18 was slightly higher in women with no evidence of disease compared with women with post-treatment HSIL (59.8% vs. 41.7%), it was lower than in women with post-treatment HPV (61.3%). Thus, HPV genotyping did not show any correlation with the outcome of the women included in the present study. In any case, the differences were not statistically significant, suggesting that no further conclusions could be drawn. These results are in keeping with the results of a previous study published by our group including a different subset of women. To provide further information on this topic, a new sentence has been added to the Discussion section of the revised manuscript (page 10, lines 342-350, underlined the new sentence) now reads:
Surprisingly, although the differences in the prevalence of post-treatment HSIL were evident, no differences were observed in terms of the prevalence of HPV16/18 at the end of follow-up between women vaccinated before or after treatment and non-vaccinated women (8.8% vs. 5.7% and 6.5%, respectively). This finding suggests that the benefit of vaccination in women undergoing treatment might not only rely on the prevention of acquiring new infections of the HPV genotypes included in the vaccine. In the present series, the HPV 16/18 genotype was not related to the outcome. This is in keeping with a previous study published by our group, which included a different subset of women (2).
QUESTION 6:
RESULTS: For the 12 women presenting post-treatment HSIL and detailed in Table 4, to correlate the HPV genotype detected pre-treatment and at the end of follow-up, it is necessary to know the time during follow-up post-treatment HSIL was detected and how is was managed, particularly who of the women underwent a second excision, since this can have modified the natural history.
RESPONSE:
We agree with the reviewer that time to post-treatment HSIL diagnosis and management of women who underwent a second conization is important. This information was already requested by the reviewer in questions 3 and 4. Thus, this information (the mean time to post-treatment HSIL diagnosis in vaccinated and non-vaccinated women and the management of women who underwent a second conization, including how they were considered for the analysis) has already been described in the revised version of the manuscript (see answers to questions 3 and 4).
As suggested by the reviewer, a new column has been added to Table 4 to include the time in months to diagnosis in each of the women with post-treatment HSIL.
QUESTION 7:
RESULTS: Finally, did the authors evaluate the outcome separately for CIN2 and CIN3?
RESPONSE:
As detailed in the Methods section, the histological specimens were classified as negative for SIL, low-grade (L)SIL, or HSIL according to the WHO criteria. Thus, in the present study, CIN2 and CIN3 were not evaluated separately. However, all cases were carefully evaluated and only histological lesions showing H&E criteria AND p16-stained positivity were categorized as HSIL to ensure that all HSIL lesions included were real premalignant lesions (transforming HPV infections).
QUESTION 8:
DISCUSSION: the paragraph on the margins' status and its correlation with the clinical outcome could be implemented, taking into account the findings of the paper published in BJOG 2020;127:377-387.
RESPONSE:
Following the reviewer’s recommendation, the new reference has been added to the paragraph on margins status and risk of persistence/recurrence after treatment for histological HSIL in the Discussion section (page 10, lines 335-337, underlined the new reference):
The prevalence of positive margins in the surgical specimen was 38.2% in the present study. Previous studies performed in non-vaccinated women have shown that positive margins represent a risk factor for developing persistent/recurrent disease (34, 35)
New reference 34: Long-term predictors of residual or recurrent cervical intraepithelial neoplasia 2-3 after treatment with a large loop excision of the transformation zone: a retrospective study. M-E Fernández-Montolí, S Tous , G Medina, M Castellarnau, A García-Tejedor, S de Sanjosé. BJOG. 2020.127(3):377-387
QUESTION 9:
ABSTRACT: the concluding sentence declaring that "...HPV vaccination before treatment reduces the prevalence..." overstates the results, that are not reported in the abstract itself; I would suggest to give more room to this aspect, reporting that the difference between pre- and post-treatment vaccine administration is statistically not significant, although it is suggestive of better protection conferred by pre-treatment vaccination.
RESPONSE:
Following the reviewer’s suggestions we have modulated the conclusion of the manuscript to avoid overstating the results. The conclusions in the revised version (page 10-11, lines 377-385, crossed out the deleted text and underlined the newly added sentences) now read:
The present series provides further evidence confirming the reduction of post-treatment HSIL in patients undergoing cervical treatment who receive HPV vaccination. Moreover, although the differences between the two strategies of vaccine administration (pre- and post-treatment) did not reach statistical significance, the results of the present series suggest that pre-treatment vaccination confers better protection. Our study shows that HPV vaccination before excisional treatment reduces the prevalence of post-treatment HSIL. Additionally, we found that vaccination might offer benefits even in women with persistent HPV after treatment. Further analysis involving a larger number of patients is needed to confirm these answers to the unsolved questions regarding the time of vaccination and its effect on treated women with persistent HPV infection.

Reviewer 2 Report
This is an interesting study, and this article is well constructed, clear and complete in my opinion. I just have two comments in the PDF for the method and results.

Author Response
QUESTION 1: A sample size paragraph is missing in this article, even if it is a sample of convenience that has been chosen over a period of time, it must be explained.
RESPONSE:
Following the reviewer’s suggestions we have included a sample size paragraph in the 2.8. section (Data Analysis).
The revised version (page 4, lines 178-181, underlined the new sentence) now reads:
Data were analyzed with SPSS software (version 28.0.0.0, SPSS, Inc, Chicago, IL, USA). A sample size calculation was not performed. However, to achieve the highest sample size possible, all women undergoing excisional treatment in our institution between July 2016 and December 2019 were included in the study.
QUESTION 2: I wonder if for this table which presents the associations, a multivariate analysis could have been carried out giving the results not only in the form of % but in the form of odds ratio which would be epidemiologically more interesting.
RESPONSE:
We agree with the reviewer that a multivariate analysis reporting the odds ratio would be interesting. Unfortunately, no women with negative pre-treatment HPV test developed HSIL after treatment. Similarly, none of the women showing no lesion in the surgical specimen developed HSIL after treatment. As a consequence, the univariate and the multivariate analysis cannot be calculated for these variables (pre-treatment HPV genotype and diagnosis of the surgical specimen). Thus, presenting the data as multivariate analysis would imply missing very relevant information of the study.
Moreover, the table provides information on all the clinical outcomes (post-treatment HSIL, post-treatment HPV and no evidence of disease at the end of follow-up) and not only about the risk of post-treatment HSIL. Thus, we believe that it provides a wider overview of the final outcomes. In conclusion, we consider that the information provided in table XX should be shown as it was in the original version of the manuscript.

Round 2
Reviewer 1 Report
The authors have adequately responded to my comments, but two modifications are necessary to complete the revision:
1-the inclusion in paragraph 3.1 (Results) of the histological diagnoses done on the excised lesions;
2-to delete in Table 3 the column regarding the "Post-treatment HSIL (n=12)", since this group of women includes the nine women who underwent a second conization and were therefore excluded from further analyses (as stated in page 4, lines 180-181).
Author Response
Two modifications are necessary to complete the revision:
QUESTION 1:
RESULTS: the inclusion in paragraph 3.1 (Results) of the histological diagnoses done on the excised lesions
RESPONSE:
Following the reviewer’s request, we have added to the text the histological diagnosis of the excised lesions, which was only shown in the table 1.
The new version of the manuscript (page 5, lines 214-216, underlined the new information) now reads:
“Table 1 shows the baseline characteristics of the two groups of women (vaccinated and non-vaccinated women), as well as the final diagnoses of the surgical specimens. Three hundred seventeen (79.6%) women showed an HSIL lesion in the surgical specimen, 40 (10.1%) women had LSIL and 41 (10.3%) did not show any lesion.”
QUESTION 2:
RESULTS: to delete in Table 3 the column regarding the "Post-treatment HSIL (n=12)", since this group of women includes the nine women who underwent a second conization and were therefore excluded from further analyses (as stated in page 4, lines 180-181).
RESPONSE:
All women with histologically confirmed HSIL after treatment were considered as having developed post-treatment HSIL. As post-treatment HSIL was one of main endpoints of the study for patients with persistent/recurrent disease, the follow-up of the nine women who underwent a second excisional treatment after this second procedure was disregarded for the purposes of the study. We believe that this information should, nonetheless, be presented. Thus, we consider that the data provided in table 3 should be shown as it was in the original version of the manuscript.
Nevertheless, we agree with the reviewer that the data management of these women should be better defined in the Methods section (2.8. Data Analysis) to avoid misunderstanding. Thus, we have modified section 2.8 to clarify this issue. The revised version (page 4, lines 181-185, underlined the new sentence and crossed out the deleted text) now reads:
“Women undergoing a second conization for histological HSIL during follow-up were considered as post-treatment HSIL being excluded from further analysis. All women with histologically confirmed HSIL after treatment were considered as having developed post-treatment HSIL. As post-treatment HSIL was one of the main outcomes of the study for patients with persistent/recurrent disease, the follow-up of the women undergoing a new excisional treatment after the second procedure was disregarded for the purposes of the study.”
